# Width-Based Lookaheads Augmented with Base Policies for Stochastic Shortest Paths

**Stefan O'Toole, Miquel Ramirez, Nir Lipovetzky, Adrian Pearce**
The University of Melbourne, Melbourne, Australia
stefan@student.unimelb.edu.au, {miquel.ramirez, nir.lipovetzky, adrianrp}@unimelb.edu.au

## Abstract

Sequential decision problems for real-world applications often need to be solved in real-time, requiring algorithms to perform well with a restricted computational budget. *Width-based* lookaheads have shown state-of-the-art performance in classical planning problems as well as over the Atari games with tight budgets. In this work we investigate width-based lookaheads over Stochastic Shortest paths (SSP). We analyse why width-based algorithms perform poorly over SSP problems, and overcome these pitfalls proposing a method to estimate costs-to-go. We formalize width-based lookaheads as an instance of the *rollout* algorithm, give a definition of width for SSP problems and explain its sample complexity. Our experimental results over a variety of SSP benchmarks show the algorithm to outperform other state-of-the-art *rollout* algorithms such as UCT and RTDP.

Keywords: width-based planning, finite-horizon MDPs, rollout algorithm, base policies

## Introduction

Model-based lookahead algorithms provide the ability to autonomously solve a large variety of sequential decision making problems. Lookaheads search for solutions by considering sequences of actions that can be made from the current state up to a certain time into the future. For real-world applications decisions often need to be computed in real-time, requiring algorithms to perform with a restricted computational budget. Limiting search in this way can result in considering states and trajectories which do not provide useful information. To address this, lookaheads can be augmented with heuristics that estimate costs-to-go to prioritise states and trajectories, and have been shown to perform well where computation budgets are restricted (Eyerich, Keller, and Helmert 2010).

This paper is concerned with Stochastic Shortest Path (SSP) problems which are often used to compare and evaluate search algorithms. We consider the *width-based* family of planning algorithms, first introduced by Lipovetzky and Geffner (2012), which aim to prioritise the exploration of *novel* areas of the state space. Two width-based planners, Lipovetzky and Geffner's breadth-first search, IW(1), and the depth-first search, Rollout-IW(1) (Bandres, Bonet, and Geffner 2018), are investigated on SSP problems. We first provide the necessary background for SSP problems and

width-based algorithms, while also formalising width-based algorithms as instances of the *rollout* algorithm (Bertsekas 2017). We then show the motive to augment width-based lookaheads with cost estimates on SSP problems, define the width of SSP problems and propose a novel width-based algorithm that estimates costs-to-go by simulating a general base policy. Our experimental study shows that the algorithm compares favourably to the original Rollout-IW(1) algorithm and to other state-of-the-art instances of the *rollout* algorithm.

## Optimal Control and Dynamic Programming

We concern ourselves with the problem of decision under stochastic uncertainty over a finite number of stages, which we characterise following closely the presentation of Bertsekas (2017). We are given a discrete-time dynamic system

$$\mathbf{x}_{k+1} = f_k(\mathbf{x}_k, u_k, \mathbf{w}_k), \quad k = 0, 1, \dots, N-1 \quad (1)$$

where the *state* $\mathbf{x}_k$ is an element of a space $S_k \subset \mathbb{R}^d$, the control $u_k$ is an element of space $C_k \subset \mathbb{N}$, and the random *disturbance* $\mathbf{w}_k$ is an element of a space $D_k \subset \mathbb{R}^{m}$ [1]. The *control* $u_k$ is constrained to take values in a given non-empty subset $U(\mathbf{x}_k) \subset C_k$, which depends on the current state $\mathbf{x}_k$, so that $u_k \in U_k(\mathbf{x}_k)$ for all $\mathbf{x}_k \in S_k$ and $k$. The random disturbance $\mathbf{w}_k$ is characterised by a probability distribution $P_k(\cdot|\mathbf{x}_k, u_k)$ that may depend explicitly on $\mathbf{x}_k$ and $u_k$ but not on the values of previous disturbances $\mathbf{w}_{k-1}, \dots, \mathbf{w}_0$. We consider the class of *policies*, or *control laws*, corresponding to the sequence of functions

$$\pi = \{\mu_0, \dots, \mu_{N+1}\} \quad (2)$$

where $\mu_k$ maps states $\mathbf{x}_k$ into controls $u_k = \mu_k(\mathbf{x}_k)$ and is such that $\mu_k(\mathbf{x}_k) \in U(\mathbf{x}_k)$ for all $\mathbf{x}_k \in S_k$. Such policies will be called *admissible*. Given an initial state $x_0$ and admissible policy $\pi$, the states $\mathbf{x}_k$ and disturbances $\mathbf{w}_k$ are random variables with distributions defined through the system equation

$$\mathbf{x}_{k+1} = f_k(\mathbf{x}_k, \mu_k(\mathbf{x}_k), \mathbf{w}_k), \quad k = 0, 1, \dots, N-1 \quad (3)$$

---

[1] We define states and disturbance as elements of subsets of the reals to avoid too specific assumptions on the structure of $S_k$, $U_k$ and $D_k$.

Thus, for given functions $g_f$ (terminal cost) and $g$ the expected cost of $\pi$ starting at $\mathbf{x}_0$ is

$$J_\pi(\mathbf{x}_0) = E\left\{ g_f(\mathbf{x}_N) + \sum_{k=0}^{N-1} g(\mathbf{x}_k, \mu_k(\mathbf{x}_k), \mathbf{w}_k) \right\} \quad (4)$$

where the expectation is taken over the random variables $\mathbf{w}_k$ and $\mathbf{x}_k$. An *optimal policy* $\pi^*$ is one that minimises this cost

$$J_{\pi^*}(\mathbf{x}_0) = \min_{\pi \in \Pi} J_\pi(\mathbf{x}_0) \quad (5)$$

where $\Pi$ is the set of all admissible policies. The optimal cost $J^*(\mathbf{x}_0)$ depends on $\mathbf{x}_0$ and is equal to $J_{\pi^*}(\mathbf{x}_0)$. We will refer to $J^*$ as the *optimal cost* or *optimal value* function that assigns to each initial state $\mathbf{x}_0$ the cost $J^*(\mathbf{x}_0)$.

## Stochastic Shortest Path

We use Bertsekas' (2017) definition, that formulates Stochastic Shortest Path (SSP) problems as the class of optimal control problems where we try to minimize

$$J_\pi(\mathbf{x}_0) = \lim_{N \to \infty} E_{w_k}\left\{ \sum_{k=0}^{N-1} \alpha^k g(\mathbf{x}_k, \mu_k(\mathbf{x}_k), \mathbf{w}_k) \right\}$$

with $\alpha$ set to 1 and we assume there is a *cost-free termination state* $\mathbf{t}$ which ensures that $J_\pi(\mathbf{x}_0)$ is finite. Once the system reaches that state, it remains there at no further cost, that is, $f(\mathbf{t}, u, \mathbf{w}) = \mathbf{t}$ with probability 1 and $g(\mathbf{t}, u, \mathbf{w}) = 0$ for all $u \in U(\mathbf{t})$. We note that the optimal control problem defined at the beginning of this section is a special case where states are pairs $(\mathbf{x}_k, k)$ and all pairs $(\mathbf{x}_N, N)$ are lumped into termination state $\mathbf{t}$.

In order to guarantee termination with probability 1, we will assume that there exists an integer $m$ such that there is a positive probability that $\mathbf{t}$ will be reached in $m$ stages or less, regardless of what $\pi$ is being used and the initial state $\mathbf{x}_0$. That is, for all admissible policies and $i = 1, ..., m$ it holds

$$\rho_\pi = \max_i P\{\mathbf{x}_m \neq \mathbf{t} \mid \mathbf{x}_0 = \mathbf{x}_i, \pi\} < 1 \quad (6)$$

A policy $\pi$ will be *proper* if the condition above is satisfied for some $m$, and *improper* otherwise.

## The Rollout Algorithm

A particularly effective on-line approach to obtain suboptimal controls is *rollout*, where the optimal cost-to-go from current state $\mathbf{x}_k$ is *approximated* by the cost of some suboptimal policy and a $d$-step lookahead strategy. The seminal RTDP (Barto, Bradtke, and Singh 1995) algorithm, is an instance of the rollout strategy where the lookahead is uniform, $d = 1$, and controls $\bar{\mu}(\mathbf{x}_k)$ selected at stage $k$ and for state $\mathbf{x}_k$ are those that attain the minimum

$$\min_{u_k \in U(\mathbf{x}_k)} E\left\{ g_k(\mathbf{x}_k, u_k, \mathbf{w}_k) + \tilde{J}_{k+1}(f_k(\mathbf{x}_k, u_k, \mathbf{w}_k)) \right\} \quad (7)$$

where $\tilde{J}_{k+1}$ is an approximation on the optimal cost-to-go $J_{k+1}^*$. If the approximation is from below, we will refer to it as a *base heuristic*, and can either be problem specific (Eyerich,

Keller, and Helmert 2010), domain independent (Bonet and Geffner 2003; Yoon, Fern, and Givan 2007) or *learnt* from interacting with a simulator (Mnih et al. 2015). Alternatively, $\tilde{J}_{k+1}$ can be defined as approximating the cost-to-go of a given suboptimal policy $\pi$, referred to as a *base policy*, where estimates are obtained via *simulation* (Rubinstein and Kroese 2017). We will denote the resulting estimate of cost-to-go as $H_k(\mathbf{x}_k)$[2]. The result of combining the lookahead strategy and the base policy or heuristic is the *rollout policy*, $\bar{\pi} \{\bar{\mu}_0, \bar{\mu}_1, ..., \bar{\mu}_{N-1}\}$ with associated cost $\bar{J}(\mathbf{x}_k)$. Such policies have the property that for all $\mathbf{x}_k$ and $k$

$$\bar{J}_k(\mathbf{x}_k) \leq H_k(\mathbf{x}_k) \quad (8)$$

when $H_k$ is approximating from above the cost-to-go of a policy, as shown by Bertsekas (2017) from the DP algorithm that defines the costs of both the base and the rollout policy. To compute at time $k$ the rollout control $\bar{\mu}(\mathbf{x}_k)$, we compute and minimize over the values of the $Q$-factors of state and control pairs $(\mathbf{x}_l, u_l)$,

$$Q_l(\mathbf{x}_l, u_l) = E\left\{ g_l(\mathbf{x}_l, u_l, \mathbf{w}_l) + Q_{l+1}(f_l(\mathbf{x}_l, u_l, \mathbf{w}_l)) \right\} \quad (9)$$

for admissible controls $u_l \in U(\mathbf{x}_l)$, $l = k + i$, with $i = 0, ..., d - 1$, and

$$Q_l(\mathbf{x}_l) = E\{H_l(\mathbf{x}_l)\} \quad (10)$$

for $l = k + d$. In this paper we make a number of assumptions to ensure the viability of lookaheads with $d > 1$. We will assume that we can *simulate* the system in Eq. 3 under the base policy, so we can generate sample system trajectories and corresponding costs consistent with probabilistic data of the problem. We further assume that we can reset the simulator of the system to an arbitrary state. Performing the simulation and calculating the rollout control still needs to be possible within the real-time constraints of the application, which is challenging as the number of $Q$-factors to estimate and minimizations to perform in Equations 9-10 is exponential on the average number of controls available per stage and $d$, the maximum depth of the lookahead. We avoid the blowup of the size of the lookahead by cutting the recursion in Equation 9 and replacing the right hand side by that of Equation 10. As detailed in the next section, we will do this when reaching states $\mathbf{x}_l$ that are deemed not to be *novel* according to the notion of structural *width* by Lipovetzky and Geffner (2012). This results in a selective strategy alternative to the upper confidence bounds (Auer, Cesa-Bianchi, and Fischer 2002) used in popular instances of Monte-Carlo Tree Search (MCTS) algorithms like Kocsis and Szepesvari's (2006) UCT, that also are instances of the rollout algorithm (Bertsekas 2017).

## Width-Based Lookaheads

We instantiate the *rollout* algorithm with an $l$-step, depth-selective *lookahead* policy using *Width-based Search* (Lipovetzky and Geffner 2012). These algorithms both focus the lookahead and have good

---

[2]We use the subindex $k$ to emphasize that the result of simulating a policy depends on the time step.

any-time behaviour. When it comes to prioritisation of expanding states, width-based methods select first states with novel valuations of features defined over the states (Lipovetzky, Ramirez, and Geffner 2015; Geffner and Geffner 2015). The most basic width-based search algorithm is $IW(1)$, a plain *breadth-first search*, guaranteed to run in *linear time and space* as it only expands *novel* states. A state $\mathbf{x}_l$ is *novel* if and only if it encounters a state variable [3] $x^i \subset \mathbb{R}$, whose value $v \in D(x^i)$, where $D(x^i)$ is the domain of variable $x^i$, has not been seen before in the current search. Note that novel states are independent of the objective function used, as the estimated cost-to-go $J$ is not used to define the novelty of the states. IW(1) has recently been integrated as an instance of a *rollout* algorithm, and has been shown to perform well with respect to learning approaches with almost real-time computation budgets over the Atari games (Bandres, Bonet, and Geffner 2018).

## Depth-First Width-Based Rollout

The breadth-first search strategy underlying IW(1) ensures a state variable $x^i$ is seen for the first time through the shortest sequence of control steps, i.e. the shortest path assuming uniform costs $g(\mathbf{x}, u, \mathbf{w})$.[4] On the other hand, *depth-first* rollout algorithms cannot guarantee this property in general. Rollout IW (RIW) changes the underlying search of IW into a depth-first rollout. In order to ensure that RIW(1) considers a state to be novel *iff* it reaches at least one value of a state variable $x_l^i$ through a shortest path, we need to adapt the definition of novelty. Intuitively, we need to define a set of state features to emulate the property of the breadth-first search strategy. Let $d(x^i, v)$ be the best upper bound known so far on the shortest path to reach each value $v \in D(x^i)$ of a state variable from the root state $\mathbf{x}_k$. Initially $d(x^i, v) = N$ for all state variables, where $N$ is the horizon which is the maximum search depth allowed for the lookahead, thus denoting no knowledge initially. When a state $\mathbf{x}_l$ is generated, $d(x^i, v)$ is set to $l$ for all state variables where $l < d(x^i, v)$.

Since RIW(1) always starts each new rollout from the current state $\mathbf{x}_k$, in order to prove a state $\mathbf{x}_l$ to be novel we have to distinguish between $\mathbf{x}_l$ being already in the lookahead tree and $\mathbf{x}_l$ being new. If $\mathbf{x}_l$ is *new* in the tree, to conclude it is novel, it is sufficient to show that there exists a state variable $x^i$ whose known shortest path value $d(x^i, v) > l$. If $\mathbf{x}_l$ is *already* in the tree, we have to prove the state contains at least one state variable value $x^i$ whose shortest path is $l = d(x^i, v)$, i.e. state $\mathbf{x}_l$ is still novel and on the shortest path to $x^i$. Otherwise the rollout is terminated.

In order to ensure the termination of RIW(1), non-novel states are marked with a *solved* label. The label is back-propagated from a state $\mathbf{x}_{l+1}$ to $\mathbf{x}_l$ if all the admissible control inputs $u \in U(\mathbf{x}_l)$ yield states $\mathbf{x}_{l+1} = f_l(\mathbf{x}_l, u, \mathbf{w}_l)$ already labeled as *solved*. RIW(1) terminates once the root state is labeled as *solved* (Bandres, Bonet, and Geffner 2018). Non-novel states $\mathbf{x}_l$ are treated as terminals and their cost-to-go is

---

[3] In order to use the notion of novelty, we assume state spaces $S$ to be stationary.

[4] This can easily be generalized to non-uniform costs by using Dijkstra's algorithm instead.

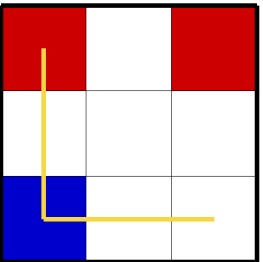

Figure 1: 3x3 `GridWorld` problem in which the blue square is the agent's initial position and the red squares show two goal locations. The yellow lines represent two action trajectories the agent can perform from the initial state.

set to $0$. This can induce a bias towards non-novel states rather than true terminal states. In the next section we investigate how to overcome the ill-behaviour of RIW(1) when a state $x_l$ is non-novel. We discuss the importance of estimating upper-bounds on the cost-to-go $H_l(\mathbf{x}_l)$ instead of assigning termination costs. This turns out to be essential for RIW(1) over SSPs.

## Width-Based Lookaheads on SSPs

Despite the successes of *width-based* algorithms on a variety of domains including the Atari-2600 games (Lipovetzky, Ramirez, and Geffner 2015; Bandres, Bonet, and Geffner 2018), the algorithms, as will be shown, have poor performance on SSP problems. We illustrate this with two scenarios. First, width-based lookaheads prefer trajectories leading to non-novel states over longer ones that reach a goal. Second, and specific to depth-first width-based lookaheads, we show that useful information is ignored. We can demonstrate these scenarios using a simple SSP problem with uniform and unitary action costs, shown in Figure 1. The task is to navigate to a goal location using the least number of left, right, up or down actions. Any action that would result in the agent moving outside of the grid produces no change in its position. The features used by the width-based planners are the coordinates for the current agent position. Both IW(1) and RIW(1) algorithms, given a sufficient budget, would result in the lookahead represented by yellow lines in Figure 1. As expected, both lookaheads contain the shortest paths to make each feature of the problem true. For both IW(1) and RIW(1), we back up the costs found in the lookahead starting from terminal and non-novel states. In this instance a move down or left from the agent's initial state has no effect, thus immediately producing a non-novel state. When backing up values, down and left have an expected cost of 1, which is less than the optimal cost of 2 for up, the action that leads to the top left goal state. This prevents both IW(1) and RIW(1) from ever achieving the goal, as they keep selecting those useless actions. Furthermore, if the goal is the top right location in Figure 1, RIW(1)'s random action selection can generate a trajectory that reaches the goal. Yet, trajectories leading to the goal are pruned away, as non-novel states in later considered trajectories are treated as terminals, again resulting in the lookahead represented by the yellow lines in Figure 1.

## Novelty, Labeling and Width of SSPs

Bandres et al. (2018) introduced the algorithm RIW in the context of deterministic transition functions. In this section we discuss its properties in the context of SSPs.

The set of features used to evaluate the novelty of a state is $F = \{(v, i, d) \mid v \in D(\mathbf{x}^i)\}$ where $D(\mathbf{x}^i)$ is the domain of variable $\mathbf{x}^i$, and $d$ is a possible shortest path distance. Note that the horizon $N$ is the upper-bound of $d$. The maximum number of novel states is $O(|F|)$, as the maximum number of shortest paths for a feature $(v, i, \cdot) \in F$ is $N$. That is, in the worst case we can improve the shortest path for $(v, i, \cdot)$ by one control input at a time.

The labeling of nodes ensures the number of rollouts from the initial state in RIW(1) is at most $O(|F| \times b)$, where $b = max_{\mathbf{x}_l}|U(\mathbf{x}_l)|$ is the maximum number of applicable control variables in a state, i.e. maximum branching factor. When the labeling is applied to stochastic shortest path problems, the resulting lookahead tree is a relaxation of the original SSP, as it allows just one possible outcome of a control input. Alternatively, one can back-propagate the label *solved* to a state $\mathbf{x}_l$ iff 1) all admissible control inputs $u \in U(\mathbf{x}_l)$ have been applied resulting in states labeled as *solved*, and 2) the tree contains all the possible resulting states of each control input $u \in U(\mathbf{x}_l)$. We refer to this new strategy to back-propagate labels as $\lambda$-labeling. We denote as $\lambda$ the maximum number of states that can result from applying $u \in U(\mathbf{x}_{l-1})$ in a state $\mathbf{x}_l$. That is, $\lambda = max_{\mathbf{x},u,w}|f(\mathbf{x}, u, \mathbf{w})|$. RIW(1) with $\lambda$-labeling will terminate after at most $O(|F| \times b \times \lambda)$ rollouts.

Furthermore, we can reconcile the notion of *width* over classical planning problems (Lipovetzky and Geffner 2012) with SSPs. A terminal state $\mathbf{t}$ made of features $f \in F$ has width 1 iff there is a trajectory $\mathbf{x_0}, u_0, \ldots, u_{n-1}, \mathbf{x}_n$ for $n \leq N$ where $\mathbf{x}_n = \mathbf{t}$, such that for each $\mathbf{x}_j$ in the trajectory 1) the prefix $\mathbf{x_0}, u_0, \ldots, u_{j-1}, \mathbf{x}_j$ reaches at least one feature $f_j = (v, i, d) \in F$ where all $(v, i, d') \in F$ for $d' < d$ are unreachable, i.e., it is a shortest path possible to reach a value in $x^i$, 2) any shortest path to $f_j$ can be extended with a single control input $u$ into a shortest path for a feature $f_{j+1}$ complying with property 1) in state $\mathbf{x}_{j+1}$, and 3) the shortest path for $f_n$ is also a shortest path for termination state $\mathbf{t}$. RIW(1) with the new labeling strategy is guaranteed to reach all width 1 terminal states $\mathbf{t}$.

**Theorem 1.** *Rollout IW(1) with $\lambda$-labeling is guaranteed to reach every width 1 terminal state $\mathbf{t}$ in polynomial time in the number of features $F$ if $\lambda = 1$.*

If $\lambda = \infty$, RIW(1) will not propagate any *solved* label, and terminate only when the computational budget is exhausted.

For simplicity, we assumed shortest paths are equivalent to the shortest sequence of control inputs. To generalize to positive non-uniform costs, the distance $d$ in the features should keep track of the cost of a path $\sum_i g(\mathbf{x}_i, u_i, \mathbf{w}_i)$ instead of its length, and the horizon be applied to the cost of the path.

## Cost-to-go Approximation

The most successful methods for obtaining cost-to-go approximations have revolved around the idea of running a number of Monte Carlo simulations of a suboptimal base policy $\pi$ (Ginsberg 1999; Coulom 2006). This amounts to generating a given number of samples for the expression minimized in Equation 7 starting from the states $\mathbf{x}_l$ over the set of admissible controls $u_l \in U(\mathbf{x}_l)$ in Equation 10, averaging the observed costs. Three main drawbacks (Bertsekas 2017) follow from this strategy. First, the costs associated with the generated trajectories may be wildly overestimating $J^*(\mathbf{x}_l)$ yet such trajectories can be very rare events for the given policy. Second, some of the controls $u_l$ may be clearly dominated by the rest, not warranting the same level of sampling effort. Third, initially promising controls may turn out to be quite bad later on. MCTS algorithms aim at combining lookaheads with stochastic simulations of policies and aim at trading off computational economy with a small risk of degrading performance. We add two new methods to the MCTS family, by combining the multi-step, width-based lookahead strategy discussed in the previous section with two simulation-based cost-to-go approximations subject to a *computational budget* that limits the number of states visited by both the lookahead and base policy simulation.

## Width-based Lookaheads with Random Walks

The first method, which we call RIW-RW, uses as the base policy a *random walk*, a stochastic policy that assigns the same probability to each of the controls $u$ admissible for state $\mathbf{x}$, and is generally regarded as the default choice when no domain knowledge is readily available. A rolling horizon $H$ is set for the rollout algorithm that follows from combining the RIW(1) lookahead with the simulation of random walks. The maximal length of the latter is set to $H - l$, where $l$ is the depth of the non-novel state. Both simulations and the unrolling of the lookahead are interrupted if the computational budget is exhausted. While this can result in trajectories sometimes falling short from a terminal, it keeps a lid on the possibility of obtaining extremely long trajectories that eat into the computational budget allowed and preclude from further extending the lookahead or sampling other potentially more useful leaf states $\mathbf{x}_l$.

## Worst-Case Estimates of Rollout Costs

One of the most striking properties of rollout algorithms is the *cost improvement* property in Equation 8, suggesting that upper bounds on costs-to-go can be used effectively to approximate the optimal costs $J^*$. Inspired by this, the second width-based MCTS method we discuss leverages the sampling techniques known as *stochastic enumeration* (SE) (Rubinstein and Kroese 2017) to obtain an *unbiased estimator* for upper bounds on costs-to-go, or in other words, estimates the maximal costs a stochastic rollout algorithm with a large depth lookahead can incur.

SE methods are inspired by a classic algorithm by D. E. Knuth to estimate the maximum search effort by backtracking search (1975). Knuth's algorithm estimates the total cost of a tree $T$ with root $u$ keeping track of two quantities, $C$ the estimate of the total cost, and $D$ the expectation on the number of nodes in $T$ at any given level of the tree, and the number of terminal nodes once the algorithm terminates. Starting with the root vertex $u$ and $D \leftarrow 1$, the algorithm proceeds by updating $D$ to be $D \leftarrow |\mathcal{S}(u)|D$ and choosing

randomly a vertex $v$ from the set of successors $\mathcal{S}(u)$ of $u$. The estimate $C$ is then updated $C \leftarrow C + c(u,v)D$ using the cost of the edge between vertices $u$ and $v$. These steps are then iterated until a vertex $v'$ is selected s.t. $\mathcal{S}(v') = 0$. We observe that Knuth's $C$ quantity would correspond to the worst-case cost-to-go $\bar{J}(\mathbf{x})_k$ of a rollout algorithm using a lookahead strategy with $d$ set to the rolling horizon $H$ and the trivial base heuristic that assigns 0 to every leaf state. Furthermore, we assume that the algorithm either does not find any terminals within the limits imposed by the computational budget assigned, or if it finds one such state, it is too the very last one being visited. Lookaheads define trees over states connected by controls, edge costs $c(u,v)$ correspond directly with realisations of the random variable $g(\mathbf{x}, u, \mathbf{w})$ and the set of successors $\mathcal{S}(v)$ of a vertex corresponds with the set of admissible controls $U(\mathbf{x})$. While Knuth's algorithm estimates are an unbiased estimator, the variance of this estimator can be exponential on the horizon, as the accuracy of the estimator lies on the assumption that costs are evenly distributed throughout the tree (Rubinstein and Kroese 2017). In the experiments discussed next, we use Knuth's algorithm directly to provide $H_k(\mathbf{x}_k)$, adding the stopping conditions to enforce the computational budget and limiting the length of trajectories to $H - l$ as above. In comparison with simulating the random walk policy, the only overhead incurred is keeping up-to-date quantities $C$ and $D$ with two multiplications and an addition.

## Experimental Study

### Domains

To evaluate the different methods we use a number of `GridWorld` (Sutton and Barto 2018) domains, an instance of a SSP problem. The goal in `GridWorld` is to move from an initial position in a grid to a goal position. In each state 4 actions are available: to move up, down, left or right. Any action that causes a move outside of the grid results in no change to the agent's position. Actions have a cost of 1, with the exception of actions that result in reaching the goal state, that have a cost of 0. The complexity of `GridWorld` can be scaled through the size of the grid and the location and number of goals. `GridWorld` also allows for extensions, which we use to have domains with a stationary goal, moving goals, obstacles and partial observability. For each instance of the `GridWorld` domain we have a $d_0 \times d_1$ grid, and the state is the current location of the agent, $\mathbf{x} = (a_0, a_1)$ where $a_i$ is the agent's position in dimension $i$. The transition function is formalised as

$$\mathbf{x}_{k+1} = \mathbf{x}_k + \mathbf{ef}_{u_k} \quad \text{if } \mathbf{x}_k + \mathbf{ef}_{u_k} \in S_{k+1} \wedge \mathbf{x}_k \notin T_k \tag{11}$$

where, $\mathbf{ef}$ specifies the change in the agent's position for each action, $T_k \subset S_k$ is the set of goal states and $\mathbf{x}_{k+1} = \mathbf{x}_k$ where the condition in Equation 11 is not met. The cost of a transition is defined as

$$g_k(\mathbf{x}_k, u_k, \mathbf{w}_k) = 0 \quad \text{if } \mathbf{x}_{k+1} \in T_{k+1} \tag{12}$$

otherwise, $g_k(\mathbf{x}_k, u_k, \mathbf{w}_k) = 1$.

For the **stationary goal** setting we have a single goal state which is positioned in the middle of the grid by dividing

| Dim. | Alg. | Heu. | Simulator Budget | | |
|------|------|------|------|------|------|
| | | | 100 | 1000 | 10000 |
| 10 | 1Stp | Rnd. | $29.6 \pm 2.5$ | $13.5 \pm 1.6$ | $7.5 \pm 0.9$ |
| | UCT | Rnd. | $29.0 \pm 2.6$ | $17.1 \pm 2.0$ | $13.3 \pm 1.5$ |
| | RIW | NA | $39.1 \pm 2.8$ | $38.1 \pm 2.9$ | $38.4 \pm 2.9$ |
| | | Rnd. | $33.7 \pm 2.5$ | $\mathbf{6.9 \pm 0.7}$ | $\mathbf{4.7 \pm 0.4}$ |
| 20 | 1Stp | Rnd. | $89.6 \pm 3.7$ | $59.8 \pm 5.2$ | $29.6 \pm 3.1$ |
| | UCT | Rnd. | $85.2 \pm 4.3$ | $72.7 \pm 5.8$ | $45.7 \pm 4.4$ |
| | RIW | NA | $79.8 \pm 5.5$ | $79.8 \pm 5.5$ | $80.2 \pm 5.5$ |
| | | Rnd. | $88.2 \pm 3.9$ | $55.3 \pm 2.2$ | $\mathbf{10.5 \pm 0.9}$ |
| 50 | 1Stp | Rnd. | $215.2 \pm 11.5$ | $201.8 \pm 13.5$ | $177.9 \pm 13.5$ |
| | UCT | Rnd. | $220.4 \pm 10.8$ | $199.2 \pm 13.5$ | $190.6 \pm 13.9$ |
| | RIW | NA | $200.2 \pm 13.8$ | $200.2 \pm 13.8$ | $200.2 \pm 13.8$ |
| | | Rnd. | $223.2 \pm 10.4$ | $199.9 \pm 13.6$ | $\mathbf{145.5 \pm 12.9}$ |

Table 1: Average and 95% confidence interval for the cost on `GridWorld` with a *stationary* goal. Costs reported are from 200 episodes over 10 different initial states (20 episodes per initial state) of the `GridWorld` with a square grid with width and length equal to the dimension (Dim.) value. The horizon of each problem is 5 times the dimension value.

and rounding $d_0$ and $d_1$ by two. The problem setting with **moving goals**, has the set of goal states modified as follows

$$T_{k+1} = \{t_k + \delta_{t_k} \mid t_k \in T_k\} \quad \text{if } \mathbf{x}_k \notin T_k \tag{13}$$

where $\delta_{t_k}$ gives the relative change of the goal state $t_k$ for the time step $k+1$ and $T_{k+1} = T_k$ if $\mathbf{x}_k \in T_k$. We use $T_0 = \{(0, d_1 - 1), (d_0 - 1, 0)\}$, $d_0 = d_1$ and

$$\delta_{t_k} = \begin{cases} (1, -1) & \text{if } t_k = (0, d_1 - 1) \\ (-1, 1) & \text{if } t_k = (d_0 - 1, 0) \\ \delta_{t_{k-1}} & \text{otherwise} \end{cases} \tag{14}$$

Resulting in two goals starting at opposite corners of the grid moving back and forth on the same diagonal. The **obstacles** setting, uses the stationary goal, but modifies $S_k$ such that,

$$S_k = \{(s_0, s_1) \mid 0 \le s_0 < d_0, \, 0 \le s_1 < d_1\} \setminus O \tag{15}$$

where $O \subset N^2$ and is the set of obstacles, that is grid cells in which the agent is not allowed.

Having partially observable obstacles in `GridWorld` provides an instance of the stochastic **Canadian Traveller Problem** (CTP) (Papadimitriou and Yannakakis 1991). The objective in CTP is to find the shortest path between one location in a road map to another, however, there is a known probability for each road in the map that due to weather conditions the road is blocked. A road in CTP can only be observed as being blocked or unblocked by visiting a location connected to it, and once a road status is observed the status remains unchanged. In terms of the `GridWorld` problem, each grid cell has a known probability of being a member of the obstacle set, $O$. The agent can only observe cells as being obstacles or not when it is in a neighbouring cell. Once a grid cell is observed it is then known that it is either an obstacle or not for the remaining duration of the episode.

John Langford designed two MDP problems[5] described as **Reinforcement Learning (RL) "Acid"** intended to be

---

[5]https://github.com/JohnLangford/RL_acid

| Dim. | Alg. | Heu. | Simulator Budget | | |
|---|---|---|---|---|---|
| | | | 100 | 1000 | 10000 |
| 10 | 1Stp | Rnd. | $17.9 \pm 2.1$ | $10.1 \pm 1.0$ | $6.8 \pm 0.5$ |
| | UCT | Rnd. | $18.9 \pm 2.2$ | $11.0 \pm 1.3$ | $10.2 \pm 1.1$ |
| | RIW | NA | $39.8 \pm 2.7$ | $38.6 \pm 2.8$ | $38.9 \pm 2.8$ |
| | | Rnd. | $21.3 \pm 2.3$ | $\mathbf{5.7 \pm 0.5}$ | $\mathbf{4.4 \pm 0.3}$ |
| 20 | 1Stp | Rnd. | $81.5 \pm 4.2$ | $45.0 \pm 4.4$ | $25.5 \pm 2.8$ |
| | UCT | Rnd. | $81.5 \pm 4.3$ | $44.9 \pm 4.7$ | $38.6 \pm 3.7$ |
| | RIW | NA | $83.5 \pm 4.8$ | $82.6 \pm 5.0$ | $82.8 \pm 4.9$ |
| | | Rnd. | $83.4 \pm 4.2$ | $39.8 \pm 4.2$ | $\mathbf{10.8 \pm 0.7}$ |
| 50 | 1Stp | Rnd. | $230.5 \pm 7.8$ | $195.3 \pm 11.8$ | $141.5 \pm 12.1$ |
| | UCT | Rnd. | $232.7 \pm 7.6$ | $196.7 \pm 11.6$ | $175.2 \pm 11.9$ |
| | RIW | NA | $212.9 \pm 11.3$ | $215.9 \pm 10.8$ | $223.5 \pm 9.8$ |
| | | Rnd. | $236.2 \pm 6.5$ | $200.4 \pm 11.4$ | $\mathbf{110.6 \pm 11.8}$ |

Table 2: Same experimental setting as Table 1 over `GridWorld` with a *moving* goal.

difficult to solve using common RL algorithms, such as Q-learning. Langford's two problems allow two actions from every state. The state space for the problems is $S_k = \{i \,|\, 0 \le i < N\}$ where the number of states value, $N$, allows the complexity of the problem to be controlled. Langford originally specified the problems as reward-based, here we modify them to be SSP cost-based problems. Reward shaping is commonly used to make Reinforcement Learning easier by encouraging actions, through higher rewards, towards a goal state or states. The first of Langford's problems is named `Antishaping` and uses reward shaping to encourage actions away from the goal state. `Antishaping` has the transition function

$$\mathbf{x}_{k+1} = \begin{cases} \mathbf{x}_k + 1 & \text{if } u_k = 0 \wedge \mathbf{x}_k \notin T_k \\ \mathbf{x}_k - 1 & \text{if } u_k = 1 \wedge \mathbf{x}_k - 1 \ge 0 \wedge \mathbf{x}_k \notin T_k \end{cases}$$
(16)

otherwise, the state remains unchanged, $\mathbf{x}_{k+1} = \mathbf{x}_k$. The set containing the goal state is $T_k = \{N - 1\}$, which can be achieved by continuously selecting $u_k = 0$. The cost of each transition in `Antishaping` is 0.25 divided by N - $x_{k+1}$, except when $x_{k+1} = N - 1$ where the cost is 0. The problem becomes a large plateau where longer paths become more costly at larger rates. The motivation behind Langford's second problem, `Combolock`, is if many actions lead back to a start state, random exploration is inadequate. The `Combolock` problem has the transition function

$$\mathbf{x}_{k+1} = \begin{cases} \mathbf{x}_k + 1 & \text{if } u_k = \mathbf{sol}_{x_k} \wedge \mathbf{x}_k \notin T_k \\ \mathbf{x}_k & \text{if } \mathbf{x}_k \in T_k \end{cases}$$
(17)

otherwise $\mathbf{x}_{k+1}$ is equal to the initial position of 0. The goal state is $T_k = \{N - 1\}$ and $\mathbf{sol}_{x_k}$ is either 0 or 1 assigned to state $x_k$ which remains constant. For each state $x \in S$, $\mathbf{sol}_x$ has an equal chance of either being 0 or 1. The cost of each transition in `Combolock` is 1 except for the transition that leads to the terminal state $N - 1$ where the cost is 0. While common Reinforcement Learning algorithms such as Q-Learning methods will struggle to solve these domains, it is claimed by Langford that the $E^3$ (Kearns and Singh 2002) family of algorithms, whose exploration do not rely solely on random policies or reward feedback but on exploring the maximum number of states, will perform well.

| Dim. | Alg. | Heu. | Simulator Budget | | |
|---|---|---|---|---|---|
| | | | 100 | 1000 | 10000 |
| 10 | 1Stp | Man. | $38.1 \pm 2.8$ | $39.0 \pm 2.7$ | $38.8 \pm 2.7$ |
| | | Rnd. | $43.9 \pm 1.9$ | $35.9 \pm 2.5$ | $25.3 \pm 2.4$ |
| | UCT | Man. | $37.0 \pm 2.9$ | $36.4 \pm 2.9$ | $36.4 \pm 2.9$ |
| | | Rnd. | $43.8 \pm 1.9$ | $38.5 \pm 2.5$ | $25.9 \pm 1.9$ |
| | RIW | Man. | $36.4 \pm 2.9$ | $36.4 \pm 2.9$ | $36.4 \pm 2.9$ |
| | | NA | $49.8 \pm 0.4$ | $48.8 \pm 1.0$ | $49.3 \pm 0.8$ |
| | | Rnd. | $44.9 \pm 1.7$ | $34.5 \pm 2.8$ | $\mathbf{19.3 \pm 2.1}$ |
| 20 | 1Stp | Man. | $76.7 \pm 5.5$ | $77.1 \pm 5.4$ | $76.7 \pm 5.5$ |
| | | Rnd. | $97.9 \pm 1.6$ | $88.0 \pm 3.4$ | $62.7 \pm 4.8$ |
| | UCT | Man. | $78.0 \pm 5.4$ | $78.4 \pm 5.3$ | $73.4 \pm 5.7$ |
| | | Rnd. | $98.7 \pm 1.2$ | $96.4 \pm 1.9$ | $77.2 \pm 4.2$ |
| | RIW | Man. | $79.7 \pm 4.9$ | $76.7 \pm 5.5$ | $76.7 \pm 5.5$ |
| | | NA | $100.0 \pm 0.0$ | $100.0 \pm 0.0$ | $99.6 \pm 0.8$ |
| | | Rnd. | $98.5 \pm 1.3$ | $88.0 \pm 3.4$ | $\mathbf{29.3 \pm 3.1}$ |
| 50 | 1Stp | Man. | $194.6 \pm 13.4$ | $191.5 \pm 13.6$ | $196.6 \pm 13.2$ |
| | | Rnd. | $249.2 \pm 1.1$ | $244.4 \pm 3.7$ | $216.8 \pm 9.3$ |
| | UCT | Man. | $194.6 \pm 13.4$ | $195.6 \pm 13.3$ | $184.4 \pm 13.9$ |
| | | Rnd. | $249.0 \pm 1.9$ | $243.1 \pm 4.3$ | $231.6 \pm 7.9$ |
| | RIW | Man. | $208.6 \pm 10.7$ | $210.6 \pm 11.1$ | $193.5 \pm 13.4$ |
| | | NA | $250.0 \pm 0.0$ | $250.0 \pm 0.0$ | $250.0 \pm 0.0$ |
| | | Rnd. | $247.9 \pm 2.6$ | $242.9 \pm 4.3$ | $196.1 \pm 11.3$ |

Table 3: Same settings as Table 1 over `GridWorld` with *fully observable obstacles* and a *stationary* goal.

## Methodology

We evaluate the depth-first width-based *rollout* algorithm, RIW(1), with and without being augmented using base policies. $\lambda = 1$ is used for the labels back-propagation. We did not observe statistically significant changes with $\lambda = \infty$. For the `GridWorld` domain we define the features on which RIW(1) plans over as $F = \{(a, i, d) \,|\, a \in D(\mathbf{x}^i)\}$ where $d$ is the length of the control input path from the initial state, $a$ is the agent's position in the grid in dimension $i$ and $D(\mathbf{x}^i)$ is the domain of the agent's position, $a$, in dimension $i$. For `Antishaping` and `Combolock` the feature set will be $F = \{(i, d) \,|\, i \in N\}$ where $i$ is the state number the agent is in and $N$ is the number of states of the domain.

Two additional rollout algorithms are also considered, the one-step *rollout* algorithm, RTDP (Barto, Bradtke, and Singh 1995) and the multi-step, selective, *regret* minimisation, *rollout* algorithm Upper Confidence bounds applied to Trees (UCT) (Kocsis and Szepevari 2006). The exploration parameter of UCT is set to 1.0 for all experiments. For all the methods that use a base policies the maximum depth of a simulated trajectory is equal to $H - l$, where $l$ is the depth at which the simulated trajectory began and $H$ is the horizon value of the lookahead. Also, a single, as opposed to multiple, simulated trajectory for the cost-to-go approximation is used, as initial results indicated it is favourable. We also report the algorithms using a Manhattan distance heuristic for the `GridWorld` domains that use obstacles. Using the Manhattan distance for the `GridWorld` problems with obstacles provides a lower bound on the cost-to-go.

Each method on the domains is evaluated at different levels of complexity by varying the number of states. The methods are evaluated using different simulator budgets. The simulator budgets are the maximum simulator calls allowed for the evaluation at each time step. For each algorithm and domain

| Dim. | Alg. | Heu. | Simulator Budget | | |
|---|---|---|---|---|---|
| | | | 100 | 1000 | 10000 |
| 10 | 1Stp | Man. | $36.6 \pm 2.9$ | $35.0 \pm 3.0$ | $35.9 \pm 2.9$ |
| | | Rnd. | $40.4 \pm 2.1$ | $27.2 \pm 2.5$ | $15.4 \pm 1.6$ |
| | UCT | Man. | $28.4 \pm 2.9$ | $29.5 \pm 3.1$ | $18.5 \pm 2.7$ |
| | | Rnd. | $41.5 \pm 2.0$ | $36.1 \pm 2.3$ | $22.2 \pm 2.0$ |
| | RIW | Man. | $28.1 \pm 3.0$ | $28.3 \pm 3.0$ | $26.5 \pm 3.0$ |
| | | NA | $49.5 \pm 0.7$ | $49.1 \pm 0.9$ | $49.8 \pm 0.4$ |
| | | Rnd. | $43.5 \pm 1.9$ | $23.4 \pm 2.6$ | $\mathbf{11.1 \pm 1.3}$ |
| 20 | 1Stp | Man. | $71.8 \pm 5.8$ | $74.0 \pm 5.7$ | $71.9 \pm 5.8$ |
| | | Rnd. | $97.0 \pm 1.7$ | $82.3 \pm 3.9$ | $49.8 \pm 4.6$ |
| | UCT | Man. | $53.8 \pm 5.7$ | $60.9 \pm 6.2$ | $38.0 \pm 5.5$ |
| | | Rnd. | $98.0 \pm 1.3$ | $87.4 \pm 4.0$ | $63.0 \pm 4.8$ |
| | RIW | Man. | $53.5 \pm 5.6$ | $\mathbf{44.1 \pm 5.5}$ | $44.1 \pm 5.6$ |
| | | NA | $100.0 \pm 0.0$ | $99.5 \pm 1.0$ | $99.5 \pm 1.0$ |
| | | Rnd. | $97.6 \pm 1.5$ | $79.7 \pm 4.4$ | $\mathbf{20.7 \pm 1.9}$ |

Table 4: Same settings as Table 1 over `GridWorld` with *partially observable* obstacles and a *stationary* goal.

| Number of States | Alg. | Heu. | Simulator Budget | | |
|---|---|---|---|---|---|
| | | | 100 | 500 | 1000 |
| 10 | 1Stp | Rnd. | $0.6 \pm 0.1$ | $0.5 \pm 0.1$ | $0.5 \pm 0.1$ |
| | UCT | Rnd. | $0.6 \pm 0.1$ | $0.5 \pm 0.1$ | $0.5 \pm 0.1$ |
| | RIW | NA | $0.7 \pm 0.1$ | $0.7 \pm 0.1$ | $0.7 \pm 0.1$ |
| | | Rnd. | $0.5 \pm 0.1$ | $\mathbf{0.3 \pm 0.0}$ | $\mathbf{0.3 \pm 0.0}$ |
| 50 | 1Stp | Rnd. | $1.7 \pm 0.1$ | $1.2 \pm 0.1$ | $1.1 \pm 0.1$ |
| | UCT | Rnd. | $1.7 \pm 0.1$ | $1.3 \pm 0.1$ | $1.2 \pm 0.1$ |
| | RIW | NA | $\mathbf{1.1 \pm 0.0}$ | $1.1 \pm 0.0$ | $1.1 \pm 0.0$ |
| | | Rnd. | $1.7 \pm 0.1$ | $1.3 \pm 0.1$ | $1.1 \pm 0.1$ |

Table 5: Average and 95% confidence interval for the cost on `Antishaping`. Costs reported are from 200 episodes over 10 different initial states (20 episodes per initial state). The horizon of each problem is 4 times the number of states.

setting we evaluate the performance over 10 different initial states with 20 episodes per initial state, equalling a total of 200 episodes. The values reported here for each algorithm and domain setting are the average and 95% confidence interval of the costs across the 200 episodes. Each episode was run using a single AMD Opteron 63xx class CPU @ 1.8 GHz, with an approximate runtime of 0.75 seconds per 1,000 simulator calls across the different algorithm and domain settings.

Results are also presented for the Atari-2600 game `Skiing`, which is a SSP problem. We use the OpenAI gym's (Brockman et al. 2016) interface of the Arcade Learning Environment (ALE) (Bellemare et al. 2013) and use the slalom game mode of `Skiing`. In the slalom mode the aim is to ski down the course as fast as possible while going through all the gates. Once the finish line is reached, a 5 second time penalty is applied for each gate that is missed. The reward values provided by ALE for `Skiing` is the negative value of the total time taken plus any time penalties in centiseconds, which we use as a cost. We use the environment settings as described by Machado et al. (2018) with a frame skip of 5 and a probability of 0.25 of repeating the previous action sent to environment instead of the current one, which Machado et al. call sticky actions. For evaluation we use a simulator budget of 100 and partial caching as described by Bandres et al. (2018), in that we cache simulator state-action transitions, thus assuming determinism, but clear the cached transitions when executing an action in the environment. However, the lookahead tree itself is not cleared when executing an action in the environment as is done for the other domains trialed. The maximum episode length is capped at 18,000 frames with a frame skip of 5 this equals 3,600 actions. Using a simulation based cost-to-go approximation is infeasible with a simulator budget of 100 and the maximum episode length of 3,600 actions. Therefore we report the algorithms using a heuristic cost-to-go estimate, which is the the number of gates that have either been missed or are still left times the time penalty of 500 centiseconds. For the RIW(1) algorithms we use the pixel values from the current gray scaled screen at full resolution, that is 180 by 210 pixels, as features.

All experiments were run within the OpenAI gym frame-

work (Brockman et al. 2016) and the code used for the algorithms and domains is available through GitHub [6].

## Results

The different $H$ functions reported here are $H_{\mathrm{NA}} = 0$, the random policy $H_{\mathrm{Rnd}}$, and the Manhattan distance $H_{\mathrm{Man}}$. The algorithms were also evaluated using Knuth's algorithm with a different range of rollouts for the cost-to-go estimate, however, the results are not reported here as they are either statistically indifferent or dominated by the results using $H_{\mathrm{Rnd}}$ with a single rollout. Bertsekas (2017) suggests that MCTS algorithms should readily benefit from stronger algorithms to estimate costs-to-go by simulation of stochastic policies. Our experiments showed that if synergies exist these do not manifest when using off-the-shelf stochastic estimation techniques like the ones discussed by Rubinstein and Kroese (2017). Table 1, 2 and 3 report the results of the different lookahead algorithms on the `GridWorld` domain variants with a stationary goal, moving goals and obstacles respectively. For these three domains, results were also collected for a 100x100 grid, however, the results were omitted from the tables as the simulator budgets used were not sufficient to find anything meaningful.

The results on the stationary goal `GridWorld` domain shown in Table 1 provide a number of insights about the *rollout* algorithms reported. First, we can see RIW(1) benefits from using $H_{\mathrm{Rnd}}$ rather than $H_{\mathrm{NA}}$ where the larger simulator budgets are used. As the simulator budget increases, as could be expected, so does the performance of all the methods using $H_{\mathrm{Rnd}}$. On the contrary, with $H_{\mathrm{NA}}$ RIW(1)'s performance remains constant across the different budgets. The explanation for this can be found in the motivating example we gave previously in this paper with the agent preferring the shorter trajectory of driving into the boundary of the grid. Table 1 also shows that given the largest budget and $H_{\mathrm{Rnd}}$, RIW(1) statistically outperforms the other algorithms on the three domains of different size.

Table 2 for `GridWorld` with moving goals has similar patterns as the stationary goal domain in that RIW(1) with $H_{\mathrm{Rnd}}$ dominates performance for the largest budget. Also, the majority of performances for the smaller budgets, excluding RIW(1) with $H_{\mathrm{NA}}$, are statistically indifferent.

---

[6]https://github.com/miquelramirez/width-lookaheads-python

| Number of States | Alg. | Heu. | Simulator Budget | | |
|---|---|---|---|---|---|
| | | | 100 | 500 | 1000 |
| 10 | 1Stp | Rnd. | $23.4 \pm 2.5$ | $13.5 \pm 2.1$ | $10.4 \pm 1.7$ |
| | UCT | Rnd. | $23.6 \pm 2.5$ | $12.7 \pm 2.0$ | $9.6 \pm 1.6$ |
| | RIW | NA | $27.4 \pm 2.6$ | $27.0 \pm 2.6$ | $27.9 \pm 2.5$ |
| | | Rnd. | $22.9 \pm 2.2$ | $\mathbf{3.6 \pm 0.4}$ | $\mathbf{3.6 \pm 0.4}$ |
| 50 | 1Stp | Rnd. | $200.0 \pm 0.0$ | $196.1 \pm 3.8$ | $191.2 \pm 5.7$ |
| | UCT | Rnd. | $199.0 \pm 1.9$ | $196.1 \pm 3.8$ | $190.2 \pm 6.0$ |
| | RIW | NA | $200.0 \pm 0.0$ | $200.0 \pm 0.0$ | $199.0 \pm 1.9$ |
| | | Rnd. | $199.0 \pm 1.9$ | $193.1 \pm 5.0$ | $190.2 \pm 6.0$ |

Table 6: Same settings as Table 5 over `Combolock`.

GridWorld with a stationary goal and obstacles results displayed in Table 3 continues the trend of results. Using the largest budget RIW(1) with $H_{\mathrm{Rnd}}$ outperforms all methods on the 10x10 and 20x20 domains. For the 50x50 a number of results are statistically indifferent. For this domain the algorithms using $H_{\mathrm{Man}}$ as the base heuristic are also reported. While using the largest budget on the 10x10 grid $H_{\mathrm{Rnd}}$ dominates $H_{\mathrm{Man}}$, for the larger 50x50 we see $H_{\mathrm{Man}}$ dominates $H_{\mathrm{Rnd}}$ for UCT, and is competitive for the other methods.

For the smallest simulator budget on CTP reported in Table 4 using $H_{\mathrm{Man}}$ with RIW(1) and UCT are the dominate methods. For the largest simulator budget RIW(1) using $H_{\mathrm{Rnd}}$ is dominant over all other methods for both sized domains. We also see that in most cases for the two smaller budgets $H_{\mathrm{Man}}$ dominates the $H_{\mathrm{Rnd}}$ methods.

Table 5 and 6 show on the smaller 10 state domains RIW(1) with $H_{\mathrm{Rnd}}$ is statistically dominant over all other methods on `Antishaping` and `Combolock` for the 500 and 1000 simulator budgets. However, for the more complex 50 state domains, the results of all algorithms using $H_{\mathrm{Rnd}}$ are statistically indifferent. It can be observed that using $H_{\mathrm{Rnd}}$ with RIW(1) does improve its performance compared with $H_{\mathrm{NA}}$ across all the domain settings with simulator budgets of 500 and 1000, besides `Antishaping` with 50 states.

For the `Skiing` Atari-2600 game results in Table 7 $H_{\mathrm{Heu}}$ is the heuristic value based on the number of gates missed and remaining as described in the previous section. RIW(1) using $H_{\mathrm{Heu}}$ dominates all other methods. Comparing RIW(1) using $H_{\mathrm{Heu}}$ results with those reported by Machado et al. (2018), it has similar performance to the DQN algorithm (Mnih et al. 2015) after 100 million frames of training. Since the simulation budget per action we use here is equivalent to 500 frames, and given that the maximum episode duration spans 3,600 actions, RIW(1) achieves the performance in Table 7 considering only 1.8 million frames.

| Alg. | Heu. | Simulator Budget |
|---|---|---|
| | | 100 |
| 1Stp | Heu. | $16{,}524.8 \pm 396.1$ |
| UCT | Heu. | $16{,}220.5 \pm 310.0$ |
| RIW | Heu. | $\mathbf{14{,}222.2 \pm 373.9}$ |
| | NA. | $15{,}854.0 \pm 332.9$ |

Table 7: Average and 95% confidence interval for the cost on the Atari-2600 `Skiing` game over 100 episodes.

than using descriptions of game states directly as we do, AlphaZero uses a CNN to extract automatically features that describe spatial relations between game pieces. Like us, AlphaZero's lookahead uses a stochastic policy to select what paths to expand, but rather than $Q$-factors, uses estimated win probabilities to prioritise controls, and *simulates* the opponent strategy via self-play to generate successor states. Our simulators are always given and remain unchanged, rather than being dynamic as is the case for AlphaZero.

Junyent et al. (2019) have recently presented a hybrid planning and learning approach that integrates Bandres et al. (2018) rollout, with a deep neural network. Similarly to AlphaZero, they use it to both guide the search, and also to extract automatically the set of state features $F$. Interestingly, Junyent et al.'s work does not use deep neural networks to approximate costs-to-go as AlphaZero does. A significant improvement in performance over Bandres et al. original rollout algorithm is reported with policies learnt after 40 million interactions with the simulator, using an overall computational budget much bigger than the one used by Bandres et al.

## Discussion

MCTS approaches typically combine lookaheads and cost-to-go approximations, along with statistical tests to determine what are the most promising directions and focus their sampling effort. The width-based methods described in this paper do so too, but in ways which are, at first sight, orthogonal to existing strategies. It remains an area of active research to map out exactly how the width-based methods described in this paper, and those elsewhere by Junyent et al. (2019) too, provide alternatives to the limitations of existing MCTS approaches. Having said this, there is no general theory guiding the design of MCTS algorithms (Bertsekas 2017), and to avoid generating ad-hoc, problem dependent solutions involuntarily it is important to follow strict protocols that alert of potential lack of statistical significance in results, while relying on a diverse set of benchmarks that can be both easily understood, and highlight limitations of existing state-of-the-art methods and overcome them.

## Related Work

Bertsekas (2017) considers AlphaGo Zero (Silver et al. 2017) to be state-of-the-art in MCTS algorithms. It combines the reasoning over confidence intervals first introduced with UCT (Kocsis and Szepevari 2006) and the classic simulation of base policies (Ginsberg 1999), adding to both supervised learning algorithms to obtain, offline, parametric representations of costs-to-go which are efficient to evaluate. The resulting algorithm achieves super-human performance at the game of Go, long considered too hard for AI agents. Rather

## Acknowledgments

This research was supported by the Australian Government Research Training Program Scholarship provided by the Australian Commonwealth Government and the University of Melbourne. The research was also supported by the Defence Science Institute, an initiative of the State Government of Victoria and Google through a Google PhD Travel Scholarship.

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
