# OpenReview forum: "Width-Based Lookaheads Augmented with Base Policies for Stochastic Shortest Paths"
_icaps-conference.org/ICAPS/2019/Workshop/HSDIP_

### Official Review · AnonReviewer1 · 2019-03-30
**The paper is well-written and an excellent fit for the workshop.**

**Rating:** 9
**Confidence:** 2

**Review:**

The paper adapts the novelty-based IW(1) algorithm to the setting of stochastic shortest paths and shows that it often performs favorably to existing approaches.

The paper introduces interesting new ideas, motivates them adequately and provides a nice evaluation on various benchmark tasks and comparing the ideas to a reasonable choice of approaches from the literature. I lack the expertise to follow all technical details, but with my limited understanding of the area, I did not spot any flaws in the technical contribution of the paper. In my opinion, the paper is an excellent fit for the workshop. I only have minor comments for the authors:

* use single hyphens between words, i.e., finite-horizon instead of finite--horizon
* drop AAAI copyright statement by using \nocopyright
* move motivation to a later place in the paper after the concepts have been introduced?

Typos:
* This paper concerns -> This paper is concerned with
* generate --in-- the lookahead
* a++n++ expected cost
* We use --in-- Bertsekas' (2017) definition
* there exists an--d-- integer
* We --instance-- ++instantiate++ the rollout algorithm
* a++n++ l-step
* has --been recently-- ++recently been++ integrated
* Dijkstra algorithm -> Dijkstra's algorithm
* Lets denote -> We denote
* i.e. it's -> i.e., it is
* This amounts to --generate-- ++generating++
* in the previous Section->section
* The maximal length of the lat++t++er is set
* a++n++ initial position
* $H_{NA}$ -> $H_\text{NA}$, etc.
* three different sized domains -> three domains of different size
* AlphaGo Zero [...] is considered by (Bertsekas 2017) and us to be state-of-the-art -> Bertsekas (2017) considers AlphaGo Zero [...] to be the state-of-the-art ?
* sate -> set
* focus there sampling effort -> focus their sampling effort

References:
* Many page numbers are missing.
* Letter case is often wrong for names and abbreviations.

---

> ### Author Response · Authors · 2019-04-09
> **Thank you for your review**
>
> Thank you for your very encouraging review and identifying these issues which we are working to address as we write this.
>
> If you have any further questions you would want us to address we are happy to answer them.

---

### Official Review · AnonReviewer2 · 2019-04-06
**The paper misses its point**

**Rating:** 7
**Confidence:** 4

**Review:**

Brief summary of the paper:
Width-based search uses a state novelty criterion to guide search and was recently used in conjunction with depth-first based rollout algorithms. However, the underlying state label procedure results in duplicate states being preferred to terminal states and as a result the algorithm is not complete. The paper proposes two solutions to overcome this issue: first, it considers stochastic outcomes when backpropagating state labels and modifies the notion of width to allow the algorithm to terminate in the deterministic case as well. Second, it proposes the use of random walk as a cost estimator to determine the cost-to-go of a pruned state, rather than treating every pruned state as a terminal state. An empirical evaluation on variations of GridWorld problems analyses the performance of the proposed algorithms compared to classical monte-carlo search algorithms.

Summary of the review:
The paper is presumably about Stochastic Shortest Path problems and the issues of width-based search algorithms on such problems, but in reality the issue is the underlying label-based pruning technique of these algorithms when faced with duplicate states. Thus, the paper somewhat misses its point and it takes a while for the reader to understand what's going on. Nevertheless, the proposed solution approaches are still valid, although not really novel, as distance-based estimators are already used throughout many different areas of search, and the paper could do a much better job in acknowledging relevant literature in the field of factored SSPs. That being said, the empirical evaluation still shows how using sample-based heuristics yields competitive performance against UCT and RTDP on a small benchmark set, which fits in the scope of the workshop.

Detailed review:
Let me preface this review with the note that I am very familar with factored SSPs, but less versed in the field of model-free SSPs. Nevertheless, I would argue that independently of having a model, a key property of a *stochastic* shortest path problem is its non-determinism. From my perspective, the GridWorld problem is a deterministic problem, as there is always a single outcome for an action given a state. That being said, I can understand that from the point of an agent with no knowledge about the world and its transition functions the actions are non-deterministic, yet the *motivation* could do a much better job in making this explicit. Additionally, to understand the motivation the reader has to be familiar with the work by Lipovetzky and Geffner and Bandres et al., and without that knowledge the problem of the example becomes only clear 3 pages later on, when solved labeling is introduced. While I do not mind it if a paper requires additional background work to be fully understandable, the motivation should be easy to follow even for readers not familiar with the background content. Since the only reason for the poor performance of the width-based algorithms is the underlying pruning based on the state labels, this could just be introduced in the motivation.

My next issue is the problem representation in general: the paper makes the claim that the underlying issue is a result of the problem being an SSP. But as I see it, the key problem with the labeling procedure is that duplicate (or non-novel) states  are marked as solved and every solved state is treated as a terminate state with a remaining cost of 0, and not that the actions are non-deterministic. I do not see why this could not happen in a deterministic domain, which allows for a noop action. That being said, the proposed solutions are still valid, but I think the paper misses the point of the underlying problem.

The paper does well in the presentation of the background. While quite formal, the problem definition is clear and not too hard to follow. On the other hand, I found the section on the redefinition of novelty very hard to follow, and exemplifying the definitions on the example given in the motivation would be very helpful for the reader. Additionally, this section somewhat conflicts with the empirical evaluation later on: lambda is the maximum number of probabilistic outcomes, and therefore fixed (by the underlying problem), but the experiment section later on treats lambda as an argument of the algorithm.

Although the paper provides apparently the first discussion of stochastic enumeration on SSPs, I note that Knuth's algorithm is just a scaled variant of random walk if all actions have the same cost and each action is applicable in every non-terminal state, as it is the case in most of the GridWorld setups, which is the reason why there is no difference in performance. Additionally, cost estimations play an important role in many different fields of search and what I really miss is at least acknowledging literature from the field of model-based factored SSPs and MDPs. For example, Trevizan et al. (2017, 2018) show how to compute heuristics for factored SSPs. Keller and Helmert (2013) present, among others, how a labeling procedure for trial-based algorithms can greatly improve the performance of UCT. I understand that this work might not be immediately applicable to the problem presented in the paper, but its relevance is in my opinion quite obvious.

The experimental study shows how the presented algorithms and modifications can improve search and outperform UCT and RDTP algorithms. Although I would not say that plain UCT and RDTP are the current state of the art, they are still quite powerful for many domains, so this is a nice result. I also note that the domain description could just be given informally, as the formal definitions are not required later on. Why do I have to read through a formal definition just to understand that the Antishaping problem consists of a n-tiled corridor where the agent can move left and right and the goal is to go all the way to the right?

Thomas Keller, Malte Helmert:
Trial-Based Heuristic Tree Search for Finite Horizon MDPs. ICAPS 2013

Felipe W. Trevizan, Sylvie Thiébaux, Patrik Haslum:
Operator Counting Heuristics for Probabilistic Planning. IJCAI 2018: 5384-5388

Felipe W. Trevizan, Sylvie Thiébaux, Patrik Haslum:
Occupation Measure Heuristics for Probabilistic Planning. ICAPS 2017: 306-315

Minor comments:

Introduction:
- 'and has been shown to perform well' => have been shown (since lookaheads is plural)
- citation in the same sentence as above: I understand that AlphaGo is cited as an example of augmenting lookaheads with heuristics, but I think that it is not the best example for restricted computation budgets, given that there is not a fixed limit per step but rather per game and the AlphaGo paper does not discuss how it distributes the total time over the individual steps

Motivation:
'would generate in the lookahead' => would generate the lookahead (or: would result in the lookahead)

Stochastic Shortest Path:
- 'we use in Bertsekas' (2017) definition' => we use Bertsekas' (2017) definition
- 'discount factor alpha' => /alpha
- 'and assuming' => and assume
- this Section => this section (this happens multiple times throughout the paper)
- w_k is not defined (do you mean bold w_k?)
- 'exists and integer' => an integer

Rollout Algorithm:
- "we avoid the blow out" => we avoid the blowup

Width-based lookaheads:
- D(x^i) is not (yet) defined

Depth-first width-based rollout:
- "IW(1) underlying breadth-first search strategy" => The breadth-first search strategy underlying IW(1)
- The horizon is not introduced/defined

Novelty, Labeling and Width of SSPs:
- "Lets denote" => Let us denote / Let's denote
- "i.e. it's the shortest path possible" => There does not only exist a single shortest path right? Then it should be 'a shortest path possible'.
- I think property 2) of the novel width criterium is a consequence of 1): if 1) holds for prefix x_0,...,x_k and feature f_k and for prefix x_0,..., x_k+1 and feature f_k+1, then we can extend the shortest path to f_k (given by x_0, ..., x_k) by u_k to feature f_k+1
- Theorem 1: If the theorem holds only when lambda is 1 this should be part of the theorem.

Width-based lookaheads:
- the later => the latter
- What is the initialization value of D?

Experimental study:
- 'allows easy scale up of complexity' => Allows to scale up the complexity
- 'ef specify' => ef specifies
- 'Langford's two problems, allow' => remove the comma
- Lambda is strictly defined, but later on it is set as a parameter. The same holds for the features: the set of features is defined as (v,i,d) for all v in the domain of x_^i, you should not define it specifically per domain (or if you do you have to change the underlying definition).

Related Work:
- "the sate of state features" => set of state features

Discussion:
- "there sampling effort" => their sampling effort

Literature:
The proceedings entry of Bandres et al. is only AAAI instead of Proc. of AAAI

---

> ### Author Response · Authors · 2019-04-09
> **Thanks for the insightful review**
>
> Thank you for your extensive and detailed review, it is most helpful. We will take care of the editorial issues you mention (typos etc.), we appreciate very much the time to flag these to us.
>
> We are processing your feedback, and have a revised version of the paper that addresses part of your concerns and reflects the results we have been obtaining since the paper was first submitted. So, while we prepare more detailed answers we would like you to further clarify the following two issues you raised:
>
> 1/ You found that the Section "Width-Based Lookaheads" was not as clear or helpful as you wished. Could you please be more specific about the parts that you think were unclear or would benefit from a detailed background?
>
> 2/ Given that the object of our work is model-free SSP problems, where we do not have access to the transition function, neither we know the probability distribution over its outputs, as we only have access to a simulator, what would be in your opinion the works that we should definitely mention? We need to comply with constraints on paper length, so it would be helpful to give priority to works which are directly relevant to our research.
>
> Thank you very much for your review again.

---

> > ### Comment · AnonReviewer2 · 2019-04-15
> > **Response**
> >
> > I am sorry for the delayed response, but I first wanted clarification regarding my questions on the discussion above.
> >
> > > You found that the Section "Width-Based Lookaheads" was not as clear or helpful as you wished. Could you please be more specific about the parts that you think were unclear or would benefit from a detailed background?
> >
> > I guess my comments above already gave some insight on this. If you want me to be more specific let me know.
> >
> > > 2/ Given that the object of our work is model-free SSP problems, where we do not have access to the transition function, neither we know the probability distribution over its outputs, as we only have access to a simulator, what would be in your opinion the works that we should definitely mention? We need to comply with constraints on paper length, so it would be helpful to give priority to works which are directly relevant to our research.
> >
> > That is a fair question. Given that there are many works on heuristics in (probabilistic) planning I don't think refering to the work by Trevizan et al. is really necessary, not only because it only works on model-based SSPs, but also because you not only consider purely probabilistic problems. It might be worth to consider putting the approach in relation to Keller and Helmert (2013) and DP-UCT. Since the underlying transition model is known, DP-UCT allows to label nodes as solved when their optimal estimated state value is known. This is of course not possible in a model-free approach, since transition probabilities are unknown, but one might view the novelty-based pruning as an approximation of an optimal pruning approach. Though, I should point out that Keller and Helmert only consider finite-horizon MDPs and not necessarily SSPs, and there might be other reasons why both approaches are comparable. While such a discussion might be interesting, it is not necessary for the paper.

---

> ### Author Response · Authors · 2019-04-09
> **Feedback on Issues of Width-based alg. Over SSPs**
>
> The issue follows from the impact of the novelty pruning mechanism on SSPs, not the labeling. Labeling just ensures termination in the absence of a budget or time bound. If labeling is used or not, the problem of IW algorithms still persists in SSPs.
>
> Rollout IW has been shown to perform very well on the ATARI games, using *deterministic* settings. In those games, typically, the cost structure (rewards) comes with two components. A per-stage reward/cost and a terminal reward/cost. The later is by default zero with notable exceptions. SKIING is an example where all the information conveyed by the cost function to the planner is given when reaching a terminal state.
>
> By using width based lookaheads - or any kind of lookahead like that of RTDP or UCT - one introduces artificial “terminal states”, that correspond to the decision points that are pruned by the lookahead strategy. One way of interpreting our work is that we’re bridging the gap that appears when one does not have an estimation of costs-to-go baked into the reward/cost function.
>
> Why this becomes a salient feature of the behaviour of the Rollout IW lookahead? Trajectories considered by Rollout IW have a length which is a function of the novelty of states. If we don’t have terminal costs baked into rewards, we are artificially biasing towards shortest trajectories. The lookahead in both UCT and RTDP have a fixed horizon, so this implicit biasing does not happen. This boundary condition is always present on SSP problems (stochastic or deterministic). It can happen as well on other settings but these have not been identified yet.

---

> > ### Comment · AnonReviewer2 · 2019-04-10
> > **rebuttal**
> >
> > > The issue follows from the impact of the novelty pruning mechanism on SSPs, not the labeling. Labeling just ensures termination in the absence of a budget or time bound. If labeling is used or not, the problem of IW algorithms still persists in SSPs.
> >
> > I think the underlying problem here is that the word pruning is only introduced when the solved labeling of the RIW algorithm is described, and additionally the informal description of width-based search algorithms says that 'when it comes to priorisation of applicable actions, width-based methods select first those that lead to states with novel valuations of features defined over states' and that IW(1) only expands 'novel' states. In my opinion this conflicts with the example in the motivation: the left and down action result in non-novel states and given only the above description I would expect that IW(1) prefers other actions. Do you disagree?
> >
> > To clarify: in the paragraph before Width-Based Lookaheads it says that we will use Eq. 10, i.e. a heuristic or simulation estimate for the successor state instead of computing the value recursively, if the current state is not novel. In the original IW and RIW algorithms pruning was implemented by setting this estimate to 0, and therefore in a cost-based (instead of reward-based) setting these actions will be selected. Is that correct?
> >
> > > By using width based lookaheads - or any kind of lookahead like that of RTDP or UCT - one introduces artificial “terminal states”, that correspond to the decision points that are pruned by the lookahead strategy. One way of interpreting our work is that we’re bridging the gap that appears when one does not have an estimation of costs-to-go baked into the reward/cost function.
> >
> > I understand that. Another example where this problem appears is the Academic Advising domain of the latest two International Probabilistic Planning Competitions (e.g. https://ipc2018-probabilistic.bitbucket.io/) where sampling based planners without a strong heuristic also only solve the problem if they are either 'lucky' or explore the whole state-space (which is possible since they know the underlying model). In general the performance of the current probabilistic online planning systems is quite weak in this domain.

---

> > > ### Author Response · Authors · 2019-04-11
> > > **Rebuttal comments**
> > >
> > > > I think the underlying problem here is that the word pruning is only introduced when the solved labeling of the RIW algorithm is described, and additionally the informal description of width-based search algorithms says that 'when it comes to priorisation of applicable actions, width-based methods select first those that lead to states with novel valuations of features defined over states' and that IW(1) only expands 'novel' states. In my opinion this conflicts with the example in the motivation: the left and down action result in non-novel states and given only the above description I would expect that IW(1) prefers other actions. Do you disagree?
> > >
> > > IW(1) will still consider non-novel states, it will just not expand them. When reaching a non-novel state instead of expanding we treat it as a terminal and apply the cost-to-go approximation. Thanks for pointing out this section we will make the description clearer in our revised version.
> > >
> > > > To clarify: in the paragraph before Width-Based Lookaheads it says that we will use Eq. 10, i.e. a heuristic or simulation estimate for the successor state instead of computing the value recursively, if the current state is not novel. In the original IW and RIW algorithms pruning was implemented by setting this estimate to 0, and therefore in a cost-based (instead of reward-based) setting these actions will be selected. Is that correct?
> > >
> > > That is correct.
> > >
> > > > Another example where this problem appears is the Academic Advising domain of the latest two International Probabilistic Planning Competitions (e.g. https://ipc2018-probabilistic.bitbucket.io/) where sampling based planners without a strong heuristic also only solve the problem if they are either 'lucky' or explore the whole state-space (which is possible since they know the underlying model). In general the performance of the current probabilistic online planning systems is quite weak in this domain.
> > >
> > > Thanks for pointing out this domain, we are in the process of testing the algorithms on it now.

---

> > > > ### Author Response · Authors · 2019-04-13
> > > > **Follow up on Academic Advising domain**
> > > >
> > > > Unfortunately, we got stuck trying to implement tf-rddlsim with our existing framework and we cannot use JAVA. So at this point in time, we will not be able to report any results on this domain.

---

> > > > > ### Comment · AnonReviewer2 · 2019-04-15
> > > > > **Thank you**
> > > > >
> > > > > Thank you for the clarification. I did not expect that you try to get results so fast, but thank you for giving it a try. Maybe you can use the domain for future work on this topic.

---

> ### Author Response · Authors · 2019-04-09
> **Feedback on deterministic vs non-deterministic**
>
> The first three variants of GridWorld are deterministic, and still SSPs as per Bertsekas' definition. Studying deterministic SSPs here is useful experimentally to isolate issues in the algorithms.
>
> The fourth SSP is the Canadian Traveller's Problem (CTP), a partially observable deterministic shortest path problem which can be compiled into an SSP, as shown in the literature (see references Eyerich, Keller and Helmert 2010, Bonet and Geffner 2012). We will shortly also post our revised paper which includes results on the ATARI - SKIING game with sticky actions, which yield several outcomes for f(s,a).
>
> References:
> Eyerich, Patrick, Thomas Keller, and Malte Helmert. "High-quality policies for the canadian traveler's problem." Twenty-Fourth AAAI Conference on Artificial Intelligence. 2010.
>
> Bonet, Blai, and Hector Geffner. "Action selection for MDPs: Anytime AO* versus UCT." Twenty-Sixth AAAI Conference on Artificial Intelligence. 2012

---

> > ### Comment · AnonReviewer2 · 2019-04-15
> > **see above**
> >
> > Please see my response to the comment above.

---

> ### Author Response · Authors · 2019-04-09
> **Feedback on Benchmarks and State-of-the-art**
>
> Given that it’s model-free SSPs with simulators, can you elaborate on what’s the state-of-the-art if not UCT and RTDP?
>
> With respect to the benchmarks, we propose deterministic cases to understand and verify the performance of algorithms. We also use partially observable cases such as CTP, and simple problems with challenging structures. We find these domains more amenable to advance understanding, and do “functional testing”, than the IPPC classical domain Exploding Blocksworld, or the Sailing problem usually discussed on approximated dynamic programming. Note as well that we need a simulator, a piece of software that needs to be constructed for each specific domain. Hence we chose domains which were easy to code and meaningful.
>
> Most papers report results on a few domains where sometimes it is not clear how diverse they are, instead of testing on a wide variety of SSPs structures, some of them designed with adversarial intent (e.g. RL Acid).
>
> If the reviewer disagrees, we’d like to hear about which domains they suggest for us to include in future versions of this work.

---

> > ### Comment · AnonReviewer2 · 2019-04-15
> > **benchmarks and state of the art**
> >
> > I am sorry for the delayed response, but I first wanted clarification regarding my questions on the discussion below. I include the second response from below in this comment, since it fits the discussion:
> >
> > > The first three variants of GridWorld are deterministic, and still SSPs as per Bertsekas' definition. Studying deterministic SSPs here is useful experimentally to isolate issues in the algorithms.
> >
> > > With respect to the benchmarks, we propose deterministic cases to understand and verify the performance of algorithms. We also use partially observable cases such as CTP, and simple problems with challenging structures.
> >
> > Given that the underlying problem is the goal-oriented nature of SSPs, and not necessarily their probabilistic aspect, I agree with the presented benchmarks (although one could argue that the grid-based problems are not well-suited to show the behaviour of Knuth's algorithm due to their symmetric nature).
> > Still, I want to emphasize that this underlying problem should be made more explicit in the paper, even if it is only to make it more accessible for readers with a more planning oriented background.
> >
> > > Given that it’s model-free SSPs with simulators, can you elaborate on what’s the state-of-the-art if not UCT and RTDP?
> >
> > I was under the impression that while the model and state space formalisation does generally not imply any specific type of model, it still corresponds to the model formalisation used in reinforcement learning approaches on Atari games, e.g. Mnih et al. (2013). But I guess this might be worth a short discussion, since I am no expert in model free learning and not completely familiar with the related literature: could deep learning approaches be applied to your type of problems? If not, why?
> >
> > Note I commented specifically on *plain* UCT/RDTP, I am aware that MCTS is still the backbone search of many state-of-the-art reinforcement learning approaches.
> >
> > Volodymyr Mnih, Koray Kavukcuoglu, David Silver, Alex Graves, Ioannis Antonoglou, Daan Wierstra, Martin A. Riedmiller: Playing Atari with Deep Reinforcement Learning. CoRR abs/1312.5602 (2013)

---

> > > ### Author Response · Authors · 2019-04-15
> > > **Comparison with Model-Free, Deep Reinforcement Learning**
> > >
> > > > could deep learning approaches be applied to your type of problems? If not, why?
> > >
> > > They do and we compare with them, just not on these domains.
> > >
> > > We do compare with them over the ATARI games, since one key aspect, the architecture of the Neural Network used to approximate Q(a,s), is "solved" for us. We either use as a baseline RL algorithm proposed by the paper by Mnih et al you quote, or we use shallow RL over the BLOB-Prost features discussed by
> > >
> > > @inproceedings{liang2016state,
> > >   title={State of the art control of atari games using shallow reinforcement learning},
> > >   author={Liang, Yitao and Machado, Marlos C and Talvitie, Erik and Bowling, Michael},
> > >   booktitle={Proceedings of the 2016 International Conference on Autonomous Agents \& Multiagent Systems},
> > >   pages={485--493},
> > >   year={2016},
> > >   organization={International Foundation for Autonomous Agents and Multiagent Systems}
> > > }
> > >
> > > In general, if one wants good performance from Deep RL algorithms some care and thought needs to be devoted to the design of the "features" states are mapped into. This is still a mostly ad-hoc process, done on a per-domain (or even per-instance!) basis, and little or no theory (that we are aware of) guides the design of these features. Some recent work on generalised planning by Frances, Bonet and Geffner, and parallel work in approximate dynamic programming , where the problem is roughly equivalent to that of "state aggregation", may offer some insight on how to obtain domain-independent feature design algorithms.
> > >
> > > We could have totally used DQN with a generic architecture on our domains but we know for a fact that peformance of those algorithms with "vanilla" networks is very poor (as in training taking very long time to converge to anything useful).
> > >
> > > In this work we have tried quite hard to avoid to handcraft features etc. something which would have benefitted probably width-based algorithms and that the baselines cannot use. The only domain-dependent elements we have allowed ourselves to use are trivial heuristics like the Manhattan Distance, which can be used by every one of the algorithms considered.

---

> > > > ### Comment · AnonReviewer2 · 2019-04-15
> > > > **Thank you**
> > > >
> > > > Thank you for the insightful comment. I think it would be great if you could fit parts of it into the paper (but obviously space is limited).

---

### Public Comment · ~Miquel_Junyent1 · 2019-04-17
**Computational budget of Junyent et al. (2019)**

Hi, we are the authors of Deep Policies for Width-Based Planning in Pixel Domains, and we found your work very interesting. Let us clarify that our approach does not use "a computational budget much bigger than the one used by Bandres et al.". Although we were not aiming to plan in real time as Bandres et al., we actually use a budget of approximately one second per step, which not only includes the planning step but also the learning step (see Table 2 in our paper https://arxiv.org/abs/1904.07091). Since we count frames instead of seconds, there is a small variance around one second (sometimes more, sometimes less, depending on the game). Importantly, compared to their 32 seconds version, we have comparable or much better results using a budget 30 times lower. It would be nice if you could update that sentence in future versions of your paper.

---

### Meta-Review · Program_Chairs · 2019-04-25

**Recommendation:** Accept
**Confidence:** 5

**Metareview:**

Dear Authors,
thank you very much for your submission. We are happy to inform you that
we have decided to accept it and we look forward to your talk in the workshop.
Please, go over the feedback in the reviews and correct or update your papers
in time for the camera ready date (May 24).
Best regards
HSDIP organizers